# Opportunistic colonoscopy in healthy individuals: A non-trivial risk of adenoma

Xiaoliang Jin[1], Chang Cai[1], Jing Zhao[1], Liang Huang[2], Bo Jin[2], Yixin Jia[3], Bin Lyu[1]*

**1** Department of Gastroenterology, The First Affiliated Hospital of Zhejiang Chinese Medical University (Zhejiang Provincial Hospital of Chinese Medical), Hangzhou, China, **2** Department of Endoscopy Center, The First Affiliated Hospital of Zhejiang Chinese Medical University (Zhejiang Provincial Hospital of Chinese Medical), Hangzhou, China, **3** Department of Gastroenterology, Zhejiang Provincial People's Hospital, Hangzhou, China

* lvbin@medmail.com.cn

## Abstract

### Background

Colorectal cancer (CRC) is the second leading cause of cancer death worldwide. Opportunistic colonoscopy may be beneficial in reducing the incidence of CRC by detecting its precursors.

### Aim

To determine the risk of colorectal adenomas in a population who underwent opportunistic colonoscopy, and demonstrate the need for opportunistic colonoscopy.

### Methods

A questionnaire was distributed to patients who underwent colonoscopy in the First Affiliated Hospital of Zhejiang Chinese Medical University from December 2021 to January 2022. The patients were divided into two groups, the opportunistic colonoscopy group who underwent a health examination including colonoscopy without intestinal symptoms due to other diseases, and the non-opportunistic group. The risk of adenomas and influence factors were analyzed.

### Results

Patients who underwent opportunistic colonoscopy had a similar risk to the non-opportunistic group, in terms of overall polyps (40.8% *vs.* 40.5%, $P = 0.919$), adenomas (25.8% *vs.* 27.6%, $P = 0.581$), advanced adenomas (8.7% *vs.* 8.6%, $P = 0.902$) and CRC (0.6% *vs.* 1.2%, $P = 0.473$). Patients with colorectal polyps and adenomas in the opportunistic colonoscopy group were younger ($P = 0.004$). There was no difference in the detection rate of polyps between patients who underwent colonoscopy as part of a health examination and those who underwent colonoscopy for other reasons. In patients with intestinal symptoms, abnormal intestinal motility and changes in stool characteristics were frequent ($P = 0.014$).

**Data Availability Statement:** The data of this study is available at https://ngdc.cncb.ac.cn/omix/release/OMIX002819.

**Funding:** This study was supported by Collaboration of Chinese traditional and Modern Medicine in Gastric Cancer and National Natural Science Foundation of China (Program number: 81970470). The funders had no role in study design, data collection and analysis, decision to publish, or preparation of the manuscript.

**Competing interests:** The authors have declared that no competing interests exist.

## Conclusion

The risk of overall colonic polyps, advanced adenomas in healthy people undergoing opportunistic colonoscopy no less than that in the patients with intestinal symptoms, positive FOBT, abnormal tumor markers, and who accepted re-colonoscopy after polypectomy. Our study indicates that more attention should be paid to the population without intestinal symptoms, especially smokers and those older than 40 years.

## Introduction

According to the 2020 global cancer statistics, colorectal cancer (CRC) is the fifth most common cancer, same is the fifth leading cause of cancer death [1]. In China, CRC is the fifth most common cancer, with an age-standardized incidence of 17.52/100 000 and an age-standardized mortality of 7.91/100 000. The incidence of CRC in urban areas is higher than that in rural areas. Moreover, it has been increasing over the past few decades [2]. Organized CRC screening programs have been carried out in some regions, such as the Kaiser Permanente Northern California (KPNC) and the Cancer Screening Program in Urban China (CanSPUC) [3]. These achieved good results, as shown by the latest statistics, the incidence rate has decreased by 25.5% compared with the baseline, and the mortality rate has decreased by 52.4% [4, 5].

The detection of colorectal tumors by colonoscopy in those who are not included in an organized screening program is considered accidental. Some people without intestinal symptoms, and do not participate in an organized screening program. This population is huge in the whole world, and lacks the management and surveillance of CRC. A recent study shown early occult CRC has increased [6], and in 2030, it is expected that 10% of colon cancer and 22% of rectal cancer will occur in people under the age of 50 years [7].

In opportunistic colonoscopy screening, the target population includes participants who aim for health examination and patients without symptoms or abnormal laboratory test results, such as abnormal tumor markers and positive fecal immunochemical test (FIT), who are recommended for colonoscopy during medical visits. In non-opportunistic screening, the population includes patients with intestinal symptoms, abnormal laboratory tests, and after polypectomy. We hope to identify the risk for CRC in asymptomatic subjects undergoing opportunistic screening, filter out the high-risk groups in this population to reduce the resources needed for screening, and examine the value of opportunistic colonoscopy.

## Materials and methods

### Study population

The Ethics Review Committee of the First Affiliated Hospital of Zhejiang Chinese Medical University approved the study (2021-KL-173-01). Regarding informed consent, we applied the ethics committee for an exemption from informed consent, and the ethics committee agreed to our request for observational research.

All patients who underwent colonoscopy in the Endoscopy Center of the First Affiliated Hospital of Zhejiang Chinese Medical University from December 2021 to January 2022 were included. The exclusion criteria were as follows: 1) Previous surgical resection of CRC, 2) aim for colonoscopic treatment, 3) regular colonoscopy due to inflammatory bowel disease (ulcerative colitis or Crohn's disease), 4) the withdrawal time of colonoscopy was less than 6 minutes.

The eligible subjects were divided into two groups: the opportunistic colonoscopy group and the non-opportunistic group. The opportunistic colonoscopy group was defined as healthy people without intestinal symptoms. The non-opportunistic group consisted of (1) those undergoing colonoscopy due to intestinal symptoms, (2) those with bloody stools or a positive FOBT result, (3) those with abnormal tumor markers, and (4) those who accepted re-colonoscopy after polypectomy.

### Data and classification

Before colonoscopy, the subjects completed a questionnaire which included information on age, sex, height, weight, intestinal preparation mode, intestinal preparation time (first preparation time, last preparation time, time of preparation to completion of colonoscopy), intestinal symptoms (including changes in stool habits or traits, bloody stools or a positive FOBT result, abdominal pain, chronic diarrhea, chronic constipation, weight loss, abdominal mass, abnormal tumor markers, anemia), history of diabetes, history of malignant tumor, history of alcohol consumption, history of smoking, family history of CRC, whether colonoscopy was performed in the past (the results of the previous colonoscopy and whether polypectomy was performed was collected if yes). The colonoscopy performed with HQ290AZI (Olympus Optical Co., Ltd., Japan). The duration of colonoscopy (min) and the quality of intestinal preparation (Boston score) were recorded.

The withdrawal time for each colonoscopy was at least 6 min [8]. Following colonoscopy, the information on patients with polyps was recorded. According to anatomical structure, the location of the polyps was divided into the proximal colon, distal colon, and whole colon; and the number of polyps was divided into single and multiple; the pathological type of polyps was divided into hyperplastic polyps, adenomas (subdivided into tubular adenomas, villous adenomas, serrated adenomas, and hybrid adenomas) and cancer; the size of polyps was divided into <10 mm and ≥10 mm. Advanced adenomas were defined as any adenoma ≥1 cm, high-grade dysplasia, or with tubulovillous or villous histology [9]. Morphology of the polyps was divided into flat, sessile, and pedunculated.

### Statistical analysis

The study size was calculated following: N = 400*(1-P)/P, P is the previous polyp detection rate in our hospital (35%), Allowable error d = 0.1P.

Continuous variables with normal distribution were expressed as Mean±SD, compared using the T-test of independent sampler. And variables with skewed distribution were expressed as median (IQR), compared using the Wilcoxon rank-sum test. Categorical variables were expressed n (%), compared using the $\chi 2$ (Pearson's chi-squared test was used when T≥5 and N≥40, Continuous correction chi-square test was used when 1≤T<5, and N≥40) or Fisher's exact test when T<1 or N<40. Logistic stepwise regression analysis was used for multivariate analysis. $P_{(bilateral)}$ <0.05 was considered statistically significant.

## Result

### Population and detection rate

782 questionnaires were collected, of which 335 cases (42.8%) had polyps. 29 cases were excluded (Fig 1), a total of 753 cases were included in the analysis group, with a mean age of 53.3±13.5 (12, 82) years; 421 females and 332 males. The opportunistic colonoscopy group had fewer diabetic patients (4.2% *vs*. 8.6%, *P* = 0.017), as shown in Table 1.

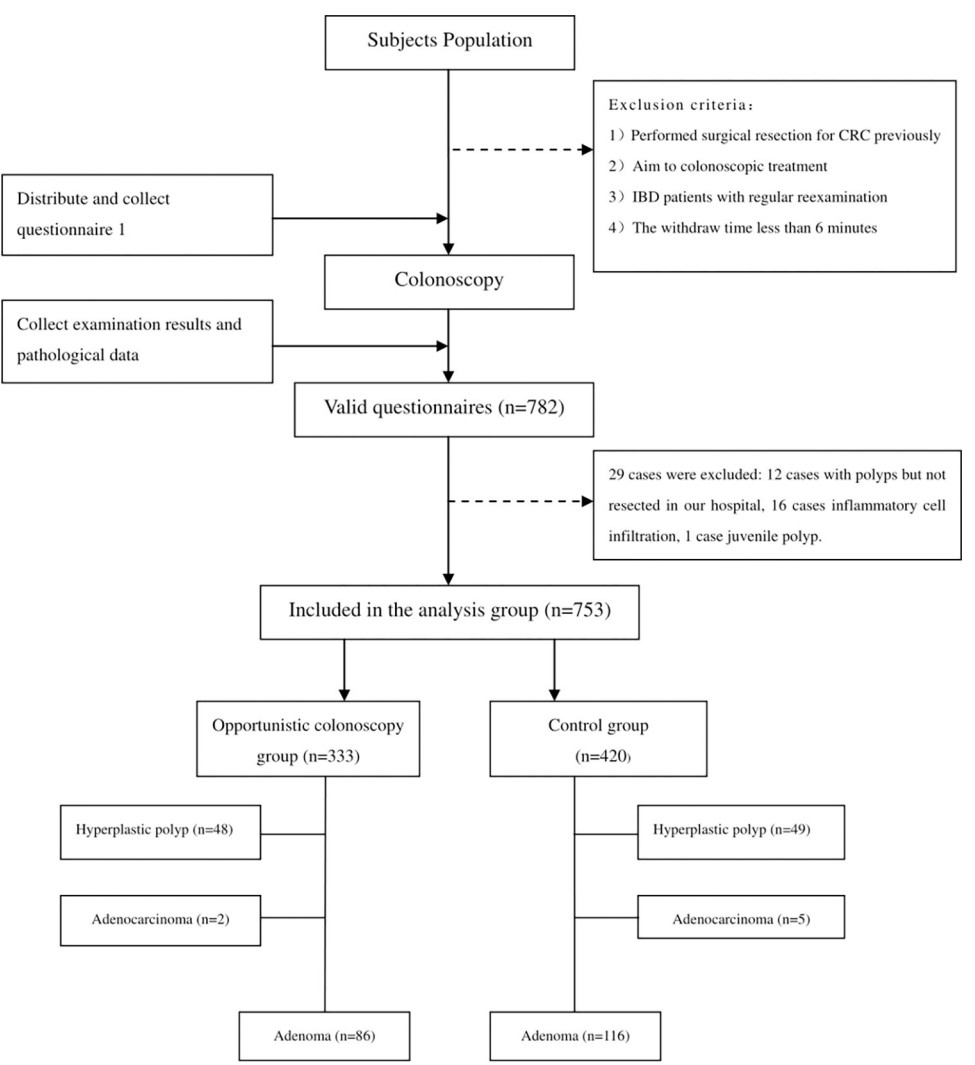

**Fig 1. Research flow chart and preliminary description of the results.** CRC: Colorectal Cancer.

There were 333 cases in the opportunistic colonoscopy group, including 136 cases of polyps (40.8%), 48 cases of hyperplastic polyps (14.4%), 86 cases of adenomas (25.8%), 31 cases of advanced adenomas (9.3%) and 2 cases of colon cancer (0.6%). In the non-opportunistic group ($n = 420$), 170 cases (40.5%) of polyps were detected, including 49 cases of hyperplastic polyps (11.7%), 116 cases of adenoma (27.6%), 38 cases of advanced adenoma (9.0%) and 5 cases of colon cancer (1.2%). There was no significant difference in the detection rate of polyps ($P = 0.919$), hyperplastic polyps ($P = 0.264$), adenomas ($P = 0.581$), advanced adenomas ($P = 0.902$), and colon cancer ($P = 0.473$) between the two groups (Table 2).

## Characteristics and risk factors

A total of 306 cases of polyps were identified, and the age of patients with polyps in the opportunistic colonoscopy group was significantly lower than that in the non-opportunistic group ($P = 0.004$). Among the 306 cases with polyps, there were 202 cases of adenomas, 86 cases in the opportunistic colonoscopy group, and 116 cases in the non-opportunistic group (63.2% *vs*. 68.2%, $P = 0.368$). The characteristics of the polyps between the two groups were compared. It

**Table 1. Demographic features of polyps by opportunistic colonoscopy.**

| | Opportunistic colonoscopy group $n$ = 333 | Non-opportunistic group $n$ = 420 | P |
|---|---|---|---|
| Age (yr) | 52.6±11.8 | 53.9±14.7 | 0.207 |
| BMI (kg/m$^2$) | 23.3±3.04 | 22.9±3.55 | 0.170 |
| Boston score | 6.0 (1.0) | 6.0 (1.0) | 0.127 |
| Diabetes (Yes) | 14 (4.2%) | 36 (8.6%) | 0.017 |
| History of malignant tumors (Yes) | 12 (3.6%) | 24 (5.7%) | 0.178 |
| Smoking history (Yes) | 93 (27.9%) | 121 (28.8%) | 0.790 |
| Alcohol intake (Yes) | 102 (30.6%) | 115 (27.4%) | 0.328 |
| Grade I relatives with CRC (Yes) | 19 (5.7%) | 39 (9.3%) | 0.067 |
| Sex | | | 0.747 |
| Female | 184 (55.3%) | 237 (56.4%) | |
| Male | 149 (44.7%) | 183 (43.6%) | |
| Intestinal preparation | | | 0.321 |
| Once | 203 (61.0%) | 241 (57.4%) | |
| Divided | 130 (39.0%) | 179 (42.6%) | |
| APCS grade | | | 0.401 |
| Low risk | 101 (30.3%) | 109 (26.0%) | |
| Middle risk | 150 (45.0%) | 198 (47.1%) | |
| High risk | 82 (24.6%) | 113 (26.9%) | |

was found that there was no significant difference in pathological type, adenoma type, number of polyps, location, size, morphology, and the proportion of advanced adenomas between the two groups (Table 3).

The risk factors for detection were analyzed further by univariate and multivariate regression. For overall polyps, Univariate analysis shown Age, Bonston scores, gender, smoking history, alcohol intake, and APCS grade were risk factors in opportunistic group, and age, BMI, gender, diabetes, smoking history, alcohol intake, APCS grade were risk factors in un-opportunistic group. For adenoma, Univariate analysis shown Age, BMI, Bonston scores, gender, smoking history, alcohol intake, and APCS grade were risk factors in opportunistic group, and age, BMI, gender, diabetes, APCS grade were risk factors in un-opportunistic group. And the independent risk factors for overall polyps and adenomas are detailed in Table 4.

To learn more about the relationship between age and colorectal polyps, we stratified the subjects by age, and it was shown that in different age groups with an interval of 10 years, both in the opportunistic colonoscopy group and the non-opportunistic group, the detection rate of colonic polyps significantly increased with increasing age; however, the incidence of colonic polyps was not statistically different between the two groups in the same age group. This was also reflected in the detection rate of adenoma. We also analyzed people with polyps in the two groups according to age and found that those with colonic polyps in the opportunistic

**Table 2. Detection rate of polyps by opportunistic colonoscopy.**

| | Opportunistic colonoscopy group $n$ = 333 | Non-opportunistic group $n$ = 420 | P |
|---|---|---|---|
| Colon polyps | 136 (40.8%) | 170 (40.5%) | 0.919 |
| Hyperplastic polyps | 48 (14.4%) | 49 (11.7%) | 0.264 |
| Adenomas | 86 (25.8%) | 116 (27.6%) | 0.581 |
| Advanced adenomas | 31 (9.3%) | 38 (9.0%) | 0.902 |
| Colorectal cancer | 2 (0.6%) | 5 (1.2%) | 0.473 |

**Table 3. Characteristics of all polyps detected by colonoscopy.**

| | Opportunistic colonoscopy group (*n* = 136) | Non-opportunistic group (*n* = 170) | *P* |
|---|---|---|---|
| Age (yr) | 56.5±9.78 | 60.1±11.9 | 0.004 |
| BMI (kg/m$^2$) | 23.6±2.88 | 23.5±3.55 | 0.745 |
| Boston score | 6.0 (1.0) | 6.0 (1.0) | 0.942 |
| Sex | | | 0.425 |
| Female | 57 (41.9%) | 79 (46.5%) | |
| Male | 79 (58.1%) | 91 (53.5%) | |
| Pathological type | | | 0.420 |
| Hyperplastic | 48 (35.3%) | 49 (28.8%) | |
| Adenoma | 86 (63.2%) | 116 (68.2%) | |
| Adenocarcinoma | 2 (1.5%) | 5 (2.9%) | |
| Number of polyp | | | 0.557 |
| Single | 90 (66.2%) | 107 (62.9%) | |
| Multiple | 46 (33.8%) | 63 (37.1%) | |
| Location | | | 0.096 |
| Proximal semicolon | 31 (22.8%) | 33 (19.4%) | |
| Distal semicolon | 95 (69.9%) | 111 (65.3%) | |
| Whole colon | 10 (7.4%) | 26 (15.3%) | |
| Size | | | 0.731 |
| <10 mm | 100 (73.5%) | 122 (71.8%) | |
| ≥10 mm | 36 (26.5%) | 48 (28.2%) | |
| Adenoma pathology | | | 0.308 |
| Tubular adenoma | 78 (90.7%) | 99 (85.3%) | |
| Villous adenoma | 2 (2.3%) | 1 (0.9%) | |
| Hybrid adenoma | 6 (7.0%) | 13 (11.2%) | |
| Serrated adenoma | 0 (0%) | 3 (2.6%) | |
| Advanced adenoma | | | 0.927 |
| No | 105 (77.2%) | 132 (77.6%) | |
| Yes | 31 (22.8%) | 38 (22.4%) | |
| Polyp morphology | | | 0.780 |
| Flat | 32 (23.5%) | 36 (21.2%) | |
| Sessile | 85 (62.5%) | 106 (62.4%) | |
| Pedunculated | 19 (14.0%) | 28 (16.5%) | |

colonoscopy group were younger, and this was also the case in patients with adenomas (Figs 2 and 3). In addition, the detection rate of overall polyps and adenomas in those over 40 years old was higher than that in younger patients (*P*<0.001, both). Also, a total of 7 cases (0.90%) with colon cancer were detected in this study.

**Table 4. Multivariate analysis of risk factors for polyps detected by opportunistic colonoscopy.**

| | Opportunistic colonoscopy group (*n* = 333) | | | | | | Non-opportunistic group (*n* = 420) | | | | | |
|---|---|---|---|---|---|---|---|---|---|---|---|---|
| | Total polyps | | | Adenoma | | | Total polyps | | | Adenoma | | |
| | P | OR | 95%CI | P | OR | 95%CI | P | OR | 95%CI | P | OR | 95%CI |
| Age (yr) | <0.001 | 1.047 | 1.025–1.070 | <0.001 | 1.054 | 1.027–1.082 | <0.001 | 1.061 | 1.043–1.079 | <0.001 | 1.071 | 1.049–1.091 |
| Sex | | | | | | | <0.001 | 2.284 | 1.485–3.513 | 0.002 | 2.155 | 1.323–3.509 |
| Smoking history | <0.001 | 3.196 | 1.908–5.352 | <0.001 | 3.926 | 2.196–7.020 | | | | | | |

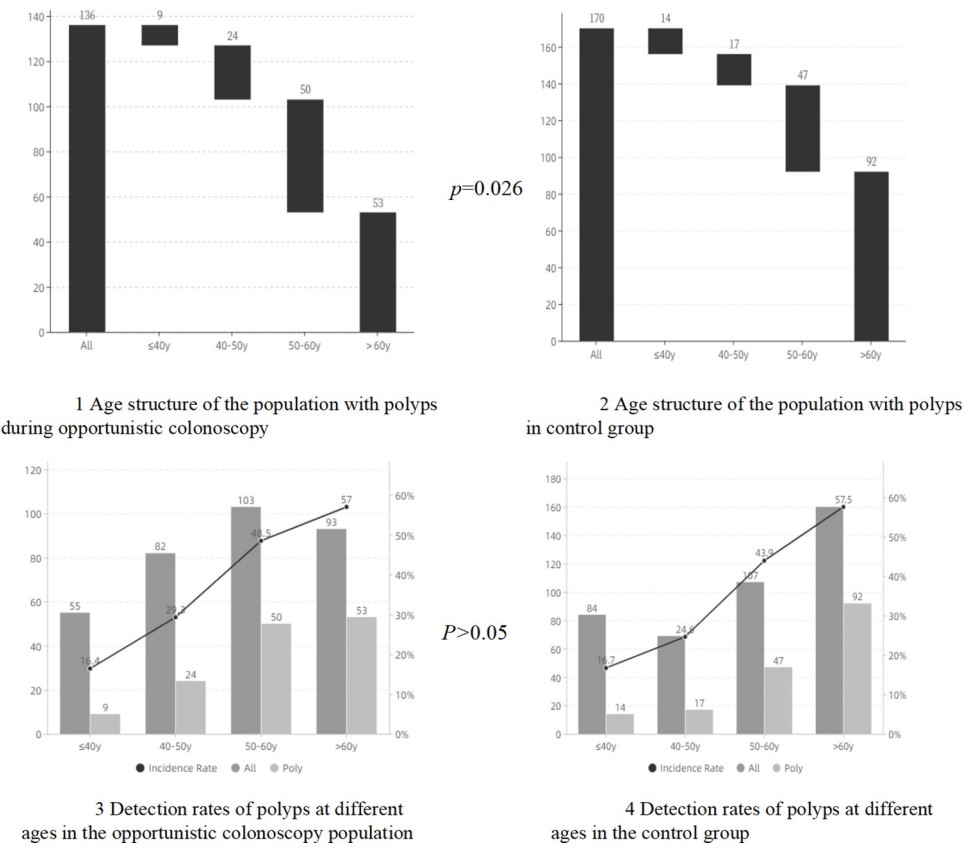

1 Age structure of the population with polyps during opportunistic colonoscopy

2 Age structure of the population with polyps in control group

3 Detection rates of polyps at different ages in the opportunistic colonoscopy population

4 Detection rates of polyps at different ages in the control group

**Fig 2. Age structure and detection rates of polyp in two groups.** Dark grey: All patients (cases). Light grey: Patients with polyps (cases). Dotted line: Incidence Rate of overall polyp (%).

## Comparison of groups in terms of the purpose of colonoscopy

According to the purpose of colonoscopy, the patients were classified as follows: re-colonoscopy after polypectomy, had a family history of CRC, patients with intestinal symptoms, and health examination. Analysis of the detection rate in patients who underwent colonoscopy for different purposes, showed that the risk of overall polyps ($P = 0.146$), hyperplastic polyps ($P = 0.242$), adenomas ($P = 0.601$), advanced adenomas ($P = 0.915$), colon cancer ($P = 0.547$) in healthy people was not lower than the other groups apparently. The population undergoing a health examination had the highest detection rate of advanced adenomas (9.3%), and the detection rate of overall polyps (40.8%) and adenoma (25.8%) was rank third.

With regard to intestinal symptoms, the detection rates of overall polyps and adenomas in people with different types of symptoms were analyzed. It was found that abnormal intestinal motility symptoms (including chronic diarrhea and constipation) and changes in stool characteristics (including bloody stool and changes in defecation habits) had higher detection rates of overall polyps (38% and 43.8%, respectively) and adenoma (28.1% and 31.3%, respectively), and the detection rate in the patients with indigestion (overall polyps: 25%, adenoma: 9.4%) and serological abnormality (overall polyps: 30%, adenoma: 23.3%) were lower.

## Discussion

CRC is a common malignant tumor of the digestive tract, and 70–90% evolve from adenomas and 10–20% evolve from serrated polyps. The incidence and mortality of CRC can be reduced

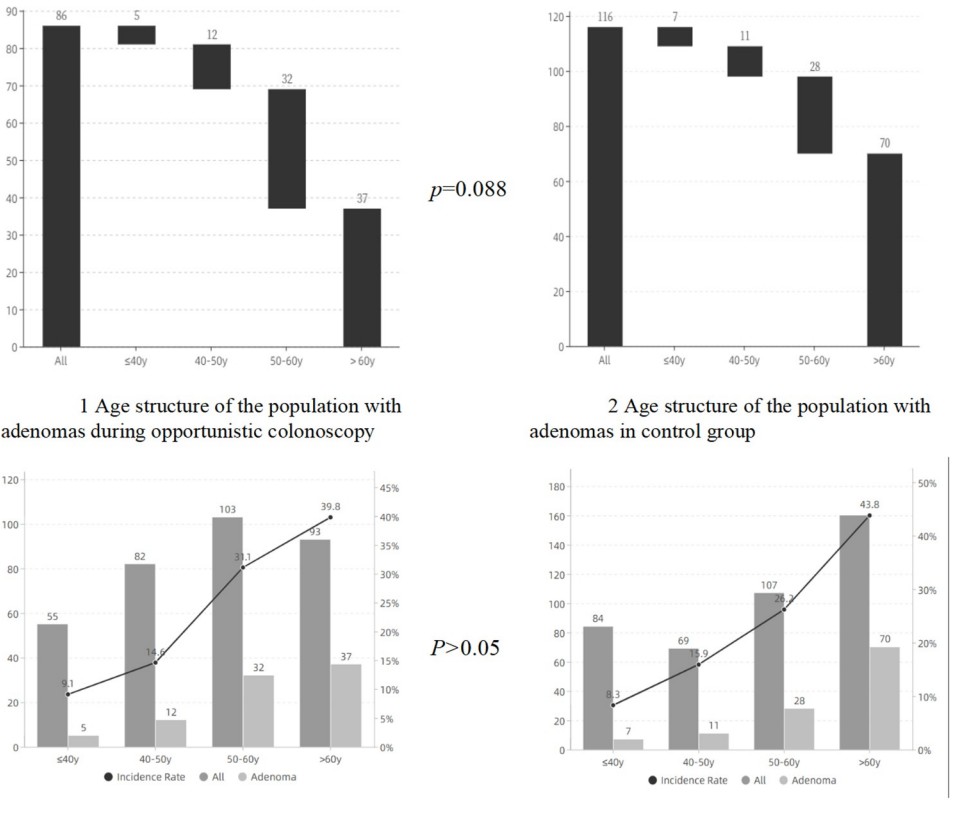

1 Age structure of the population with adenomas during opportunistic colonoscopy

2 Age structure of the population with adenomas in control group

3 Detection rates of adenomas at different ages in the opportunistic colonoscopy population

4 Detection rates of adenomas at different ages in the control group

**Fig 3. Age structure and detection rates of adenoma in two groups.** Dark grey: All patients (cases). Light grey: Patients with adenomas (cases). Dotted line: Incidence Rate of adenoma (%).

(18–26% and 22–31%, respectively) if it is found and removed before malignant transformation [10]. As a measure of screening, colonoscopy had an increased participation rate in CRC screening in the United States from 41.1% in 2000 to 83.1% in 2015 [5]. Recent studies have found that optical diagnosis under colonoscopy is close to pathological diagnosis [11], reflecting the importance of colonoscopy for screening and diagnosis. The purpose of this study was to examine the value of opportunistic colonoscopy screening.

In this study, we analyzed 753 patients who underwent colonoscopy. The overall detection rate of polyps was 40.6% (306/753), and the detection rate of adenomas was 26.8% (202/753), which was similar to that of IJspeert *et al* (the detection rate of adenomas was 29.4–32.3%; the detection rate of serrated polyps was 26.6–27.2%) [12]. The detection rate of adenomas and colon cancer was 27.8% (209/753), which was higher than that of 21.9% in a prospective cohort study in Beijing in 2016 [13]. We found that there was no significant difference between the opportunistic colonoscopy group and the non-opportunistic group in the detection rate of adenomas (25.8% *vs*. 27.6%, *P* = 0.581), advanced adenomas (9.3% *vs*. 9.0%, *P* = 0.902), and colon cancer (0.6% *vs*. 1.2%, *P* = 0.473), as shown in Table 1. When the detection rate of colon adenoma/cancer reached 1/4 in the overall population, it can be seen that patients undergoing opportunistic colonoscopy also had a high risk of colorectal adenoma/cancer, which is disadvantageous in asymptomatic people without supervision.

In screening efforts, we have found that some screening subjects forgo colonoscopy because there are no obvious signs of gastrointestinal symptoms, which our study shows is undesirable.

A previous European study [12] showed that the detection of serrated polyps (SP) and serrated polyp syndrome was not related to symptoms, and may be related to destruction of the enteric nervous system [14], changes in intestinal flora [15], polyp bleeding [16], and other reasons. The results of this study also showed that there was no statistical difference in the detection rate of colorectal polyps, adenomas, and advanced adenomas between the two groups, further suggesting that colonoscopy results could not be determined according to the presence of intestinal symptoms. According to the prediction model of CRC risk assessment established by Usher [17] and Williams [18], the efficiency of the prediction model for advanced polyps in the symptomatic population was similar to that in the asymptomatic population, suggesting that for asymptomatic people, it is biased to judge whether colonoscopy should be performed based on intestinal symptoms. This phenomenon may lead patients with symptoms to undergo colonoscopy, detect and even resect polyps in time. Approximately 40.8% of asymptomatic patients may increase their risk of developing CRC in the future, and 9.3% of subjects have a higher risk, according to our study.

To understand the risk factors for overall polyps and adenomas in those who participated in opportunistic colonoscopy, we conducted univariate and multivariate analyses (Table 3). As the APCS grade included information on sex, age, smoking history, and grade I relatives with CRC, it was not included in the multivariate logistic regression analysis. Univariate analysis showed that the high-risk group with APCS grade had a higher detection rate of overall polyps and adenomas in the opportunistic colonoscopy group and the non-opportunistic group. Multivariate analysis showed that age and smoking were independent risk factors for overall polyps and adenomas in the opportunistic colonoscopy group, while age and sex were independent risk factors for overall polyps and adenomas in the non-opportunistic group. Considering that age was an independent risk factor in both groups, we carried out a stratified analysis and found that patients with overall polyps and adenomas in the opportunistic colonoscopy group were younger (Figs 2, 3). In addition, both groups showed that the detection rate of overall polyps and adenomas increased with age. Taking 40 years old as the cutoff, a significant increase in the detection rate of colonic polyps and adenomas was observed in the population aged over 40 years (Table 3).

Regarding the detection of polyps in people who underwent colonoscopy for different purposes, analysis showed that the detection rate of polyps in those who underwent re-colonoscopy was highest, and the highest detection rate of adenomas was in those with a family history of colon cancer. What is surprising is that among the subjects examined by opportunistic colonoscopy, the detection rate of advanced adenomas was highest, and the relationship between the detection rates is shown in Fig 2. In this study, a total of 7 cases (0.90%) of CRC were detected, of which 2 cases were detected in the opportunistic colonoscopy group and 5 cases in the non-opportunistic group. Overall, our study shows the importance of opportunistic colonoscopy screening, due to the high rate of polyp and adenoma detection in its participant population. At the same time, it is unreliable to judge the need for opportunistic screening by symptoms.

This study had the following limitations: 1) this was a single-center observational cohort study, without any intervention in the subjects and the samples of advanced adenomas detected were insufficient. 2) The time interval between the subjects who underwent re-colonoscopy after polypectomy to examination was not assessed, so it is difficult to analyze the relationship between the re-detection of polyps and the time of re-colonoscopy.

## Conclusion

Our study supports 40 years as the age cut-off for screening because of its significantly higher risk of colon polyps. Also, the risk of detecting adenomas was higher among smokers.

Furthermore, it would be unwise for the patient or the doctor to forego colonoscopy during screening because of the absence of gastrointestinal symptoms or abnormal laboratory tests.

## Author Contributions

**Conceptualization:** Chang Cai, Jing Zhao, Bin Lyu.

**Data curation:** Bo Jin.

**Formal analysis:** Xiaoliang Jin.

**Investigation:** Xiaoliang Jin, Chang Cai, Yixin Jia.

**Methodology:** Xiaoliang Jin.

**Resources:** Liang Huang.

**Supervision:** Chang Cai, Jing Zhao, Bo Jin, Bin Lyu.

**Writing – original draft:** Xiaoliang Jin.

**Writing – review & editing:** Xiaoliang Jin, Bin Lyu.

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
