## [Decision Letter · Decision Letter 0]

3 Jan 2023

PONE-D-22-29921Opportunistic colonoscopy in healthy individuals: a non-trivial risk of adenomaPLOS ONE

Dear Dr. Lyu,

Thank you for submitting your manuscript to PLOS ONE. After careful consideration, we feel that it has merit but does not fully meet PLOS ONE’s publication criteria as it currently stands. Therefore, we invite you to submit a revised version of the manuscript that addresses the points raised during the review process.

We look forward to receiving your revised manuscript.

Kind regards,

Hsu-Heng Yen

Academic Editor

PLOS ONE

Journal Requirements:

"The funders had no role in study design, data collection and analysis, decision to publish, or preparation of the manuscript"

4. Please amend the manuscript submission data (via Edit Submission) to include author Yixin Jia.

Reviewers' comments:

Reviewer's Responses to Questions

**Comments to the Author**

1. Is the manuscript technically sound, and do the data support the conclusions?

Reviewer #1: Yes

Reviewer #2: Yes

2. Has the statistical analysis been performed appropriately and rigorously? 

Reviewer #1: Yes

Reviewer #2: Yes

3. Have the authors made all data underlying the findings in their manuscript fully available?

Reviewer #1: Yes

Reviewer #2: Yes

4. Is the manuscript presented in an intelligible fashion and written in standard English?

Reviewer #1: Yes

Reviewer #2: Yes

5. Review Comments to the Author

Reviewer #1: Minor queries;

1. Could you please describe what an opportunistic colonoscopy is in the introduction section?

Clearly, the patients undergoing colonoscopy without having any intestinal symptoms were called opportunistic group. It is a screening for sure.

But, what about non-opportunistic group? These patients were enrolled for colonoscopy since they had some reason for colonoscopy indeed. Please rewrite the introduction section.

Instead of giving too much details of some programs like Keiser etc, please focus on describing the opportuistic and non-opportunistic terms.

It takes time to understand what you actually meant.

2. And also, I would suggest putting all the subjects with a family history of colorectal cancer into the non-opportunistic group.

3. And tables, there is too much information and they are too caotic. Could you please give at least only one information per row? Like, give only the female percentage, or smnoker or alcohol intakers, instead saying `yes` and `no`groups both.

4. In tabel 1, please focus on giving only baseline demographics; no need to give the polyps data here. In table 2, you can gather all the polyp details.

5. Table 3 is too caotic again. Please give only the results of your regression analyses results, instead of giving all the details and differences between groups.

6. No need for table 4. Please give some small information about the data given in table 4.

7. It is interesting to see that having a family hostoy of cancer is not a risk factor. How could the authors interpret this result?

Was that expected?

8. I don't think that table 5 is necessary.

Reviewer #2: This study is interesting. According to your study, the public policy of colon cancer screen may be needed adjustment. But as your statement, the samples of the advanced adenomas were insufficient. The relative benefit in colonoscopy screen to whole population above 40 year old without any indication may be needed more studies.

6. PLOS authors have the option to publish the peer review history of their article (what does this mean?). If published, this will include your full peer review and any attached files.

Reviewer #1: **Yes: **Akif Altinbas

Reviewer #2: No

---

## [Author Response · Author response to Decision Letter 0]

21 Feb 2023

Reviewer #1: Minor queries;

1. Could you please describe what an opportunistic colonoscopy is in the introduction section? Clearly, the patients undergoing colonoscopy without having any intestinal symptoms were called opportunistic group. It is a screening for sure. But, what about non-opportunistic group? These patients were enrolled for colonoscopy since they had some reason for colonoscopy indeed. Please rewrite the introduction section. Instead of giving too much details of some programs like Keiser etc, please focus on describing the opportunistic and non-opportunistic terms. It takes time to understand what you actually meant.

Response: Thank you for the comment, In opportunistic colonoscopy screening, the target population includes participants who aim for health examination and patients without symptoms or abnormal laboratory test results, such as abnormal tumor markers and positive fecal immunochemical test, who are recommended for colonoscopy during medical visits. In non-opportunistic screening, the participants always have an established colonoscopy program, such as patients with intestinal symptoms, abnormal laboratory tests (tumor markers, positive FIT), and after polypectomy. 

In the introduction section, we have rewritten this part, focusing on the introduction of the opportunistic and un-opportunistic groups (Paragraph 3, Introduction section), thank you again.

2. And also, I would suggest putting all the subjects with a family history of colorectal cancer into the non-opportunistic group.

Response: Dear reviewer, thank you for your comments. The indications of the two groups in our study were described in comment 1. A family history of colorectal cancer is one of the risk factors for colon cancer and is an important consideration for colonoscopy screening. During health check-ups, or medical visits for other diseases, colonoscopy screening is recommended for people without gastrointestinal symptoms or abnormal laboratory tests, based on the Asia-Pacific Colorectal Cancer Screening score, if they have a history of first-degree relatives with colorectal cancer, are smokers, are male and are older than 50y. They are also the target of opportunistic screening, which includes subjects with a family history of colon cancer. Therefore, we consider that subjects with a family history of colon cancer should also be classified as opportunistic and non-opportunistic groups according to their different statuses.

3. And tables, there is too much information and they are too caotic. Could you please give at least only one information per row? Like, give only the female percentage, or smoker or alcohol intakers, instead saying `yes` and `no`groups both.

Response: Thank you for the comments, we have realized it and revised all the tables in our article, the dichotomous variables were shown in a positive item, such as smoking history, alcohol intake, colon polyps, and so on.

4. In table 1, please focus on giving only baseline demographics; no need to give the polyps data here. In table 2, you can gather all the polyp details.

Response: Thank you for the comment, we revised Table 1, and in the revised manuscript Table 1 shows the baseline demographics only, and Table 2 shows the detection rate in two groups. And describe the characteristics of all polyps detected by colonoscopy in table 3.

5. Table 3 is too caotic again. Please give only the results of your regression analyses results, instead of giving all the details and differences between groups.

Response: thank you for the comments, we delete the univariate result in table 3, and the regression analysis result was shown in the revised table 4. And we also add the significant result of univariate analyses in the Result section and rewrite this part (lines 164-172, revised manuscript with track changes)

6. No need for table 4. Please give some small information about the data given in table 4.

Response: thank you for the comments, we delete table 4, and we add some information from table 4 in the Results section (lines 203-204, revised manuscript with track changes).

7. It is interesting to see that having a family history of cancer is not a risk factor. How could the authors interpret this result?

Was that expected?

Response: A family history of colon cancer is an independent risk factor for the development of colon adenoma and cancer, as has been demonstrated in many studies. However, our study showed that a first-degree family history of colon cancer was not a risk factor in either the opportunistic or non-opportunistic colonoscopy group by univariate analysis. Meanwhile, we can see that in the opportunistic colonoscopy group, the overall polyp (57.9%) and adenoma (36.8%) detection rates were higher in those with first-degree family history than in those without a first-degree family history (39.8% total polyps and 25.2% adenomas), whereas in the non-opportunistic group, the overall polyp (38.5%) and adenoma detection rates in those with a first-degree family history ( 28.2%) were not higher than those without a first-degree family history (40.7% for total polyps and 27.6% for adenomas). We extended the family history to include, but not be limited to, first-degree relatives and performed statistical analysis with the detection rates shown in the table below

Detection rate opportunistic group non-opportunistic group

 Overall polyps adenoma Overall polyps adenoma

Grade Ⅰrelatives with CRC (+) 57.9% 36.8% 38.5% 28.2%

Grade Ⅰrelatives with CRC (-) 39.8% 25.2% 40.7% 27.6%

Family history of CRC (+) 46.6% 26.6% 40% 26%

Family history of CRC (-) 40.2% 25.7% 40.5% 27.8%

With the above data, an increase in the family history grade of CRC did raise the overall risk of polyps and adenomas, and this manifestation was more pronounced in the opportunistic group. This may be related to the non-opportunistic group being confounded with other risk factors, such as a history of colon adenoma resection, intestinal symptoms, and so on. In addition, the number of patients with a first-degree family history was only 19 and 39 in the opportunistic and non-opportunistic groups, and the sample size may be one of the reasons for the absence of statistical differences, thus not sufficient to negate the role of family history in the development of colorectal cancer.

8. I don't think that table 5 is necessary.

Response: Thank you for the comments, we delete table 5, and we add some information from table 5 in the Results section (lines 209-212, revised manuscript with track changes).

Reviewer #2: This study is interesting. According to your study, the public policy of colon cancer screen may be needed adjustment. But as your statement, the samples of the advanced adenomas were insufficient. The relative benefit in colonoscopy screen to whole population above 40 year old without any indication may be needed more studies.

Response: Thank you for your comments, which provide a complete review of our article. During screening, the problem of screening subjects refusing colonoscopy due to lack of gastrointestinal symptoms, abnormal tumour markers, etc. is often encountered. Our study supports the importance of screening and clarifies the issue: gastrointestinal symptoms are not used as one of the criteria for whether to screen or not. In line with this argument, we have made some revisions in the discussion section.

---

## [Decision Letter · Decision Letter 1]

13 Mar 2023

Opportunistic colonoscopy in healthy individuals: a non-trivial risk of adenoma

PONE-D-22-29921R1

Dear Dr. Lyu,

We’re pleased to inform you that your manuscript has been judged scientifically suitable for publication and will be formally accepted for publication once it meets all outstanding technical requirements.

Kind regards,

Hsu-Heng Yen

Academic Editor

PLOS ONE

Additional Editor Comments (optional):

Reviewers' comments:

Reviewer's Responses to Questions

**Comments to the Author**

1. If the authors have adequately addressed your comments raised in a previous round of review and you feel that this manuscript is now acceptable for publication, you may indicate that here to bypass the “Comments to the Author” section, enter your conflict of interest statement in the “Confidential to Editor” section, and submit your "Accept" recommendation.

Reviewer #1: All comments have been addressed

2. Is the manuscript technically sound, and do the data support the conclusions?

Reviewer #1: Yes

3. Has the statistical analysis been performed appropriately and rigorously? 

Reviewer #1: Yes

4. Have the authors made all data underlying the findings in their manuscript fully available?

Reviewer #1: Yes

5. Is the manuscript presented in an intelligible fashion and written in standard English?

Reviewer #1: Yes

6. Review Comments to the Author

Reviewer #1: To the Editor,

I would suggest accepting the current version of the article to publish in your journal.

BEst

7. PLOS authors have the option to publish the peer review history of their article (what does this mean?). If published, this will include your full peer review and any attached files.

Reviewer #1: **Yes: **Akif Altinbas

---

## [Editor Report · Acceptance letter]

4 Apr 2023

PONE-D-22-29921R1 

Opportunistic colonoscopy in healthy individuals: a non-trivial risk of adenoma 

Dear Dr. Lyu:

I'm pleased to inform you that your manuscript has been deemed suitable for publication in PLOS ONE. Congratulations! Your manuscript is now with our production department. 

Kind regards, 

on behalf of

Dr. Hsu-Heng Yen 

Academic Editor

PLOS ONE